# SNPP VIIRS Day Night Band: Ten Years of On-Orbit Calibration and Performance

Hongda Chen [1,*], Chengbo Sun [2], Xiaoxiong Xiong [3], Gal Sarid [1] and Junqiang Sun [1]

1   Science Systems and Applications, Inc., 10210 Greenbelt Road, Lanham, MD 20706, USA;
    gal.sarid@ssaihq.com (G.S.); junqiang.sun@ssaihq.com (J.S.)
2   Global Science and Technology, Inc., 7855 Walker Drive, Greenbelt, MD 20770, USA; chengbo.sun@gst.com
3   Sciences and Exploration Directorate, NASA/GSFC, Greenbelt, MD 20771, USA; xiaoxiong.xiong-1@nasa.gov
*   Correspondence: hongda.chen@ssaihq.com; Tel.: +1-301-867-2045; Fax: +1-301-867-6246

**Abstract:** Aboard the polar-orbiting SNPP satellite, the VIIRS instrument has been in operation since launch in October 2011. It is a visible and infrared radiometer with a unique panchromatic channel capability designated as a day-night band (DNB). This channel covers wavelengths from 0.5 to 0.9 μm and is designed with a near-constant spatial resolution for Earth observations 24 h a day. The DNB operates at 3 gain stages (low, middle, and high) to cover a large dynamic range. An onboard solar diffuser (SD) is used for calibration in the low gain stage, and to enable the derivation of gain ratios between the different stages. In this paper, we present the SNPP VIIRS DNB calibration performed by the NASA VIIRS characterization support team (VCST). The DNB calibration algorithms are described to generate the calibration coefficient look up tables (LUTs) for the latest NASA Level 1B Collection 2 products. We provide an evaluation of DNB on-orbit calibration performance. This activity supports the NASA Earth science community by delivering consistent VIIRS sensor data products via the Land Science Investigator-led Processing Systems, including the SD degradation applied for DNB calibrations in detector gain and gain ratio trending. The DNB stray light contamination and its correction are highlighted. Performance validations are presented using comparisons to the calibration methods employed by NOAA's operational Interface Data Processing Segment. Further work on stray light corrections is also discussed.

**Keywords:** VIIRS; VCST; DNB; SNPP; calibration

## 1. Introduction

As one of the main instruments onboard the Suomi-national polar-orbiting partnership (SNPP) satellite, the visible infrared imaging radiometer suite (VIIRS) is a highly capable and diverse sensor array, in terms of its available visible and infrared measurements and their derived applications. It represents an updated and improved version of the moderate resolution imaging spectroradiometer (MODIS) onboard the NASA Earth observing system (EOS) missions Aqua and Terra. The SNPP satellite was successfully launched on 28 October 2011, as the first platform in the new generation of NASA/NOAA-managed polar-orbiting satellites, extending NASA's EOS capabilities [1–3]. VIIRS's first-light observations were available as soon as 21 November 2011 and 19 January 2012, for visible and infrared images, respectively.

The VIIRS scanning mechanism uses a rotating telescope assembly (RTA) and a half-angle mirror (HAM). This setup, with both HAM sides, has a period of rotation of 1.78 s. This is applied to the cross-track scanning and produces observations of Earth scenes through the Earth view (EV) port. EV imagery is taken over a swath of approximately 3000 km in width (along-scan) and 12 km in length (along-track). This swath is produced by a range of scan angles around the nadir, which varies from +56.28° to −56.28°. The satellite's has a mean orbital altitude of 838 km, varying between 828 km near 15° north and 856 km near the South Pole. The 22 spectral bands of VIIRS cover a range between

0.410 and 12.50 μm, and are separated into 16 moderate and 5 imaging resolution bands, which are collectively designated as reflective solar bands (RSB), thermal emissive bands (TEB), and 1 panchromatic band in the visible to near-infrared range (0.5~0.9 μm). This latter band is designated as the day-night band (DNB), and was designed to have a large dynamic range [1–6]. This property of the DNB makes it sensitive to very low levels of visible light at night.

The DNB dynamic range is $3.0 \times 10^{-9}$ Wcm$^{-2}$ sr$^{-1}$ to at least $2.0 \times 10^{-2}$ Wcm$^{-2}$ sr$^{-1}$, and it is covered by 4 detector arrays, which are set at 3 different radiometric gains. They are termed as the low gain stage (LGS), the middle gain stage (MGS), and the high gain stage (HGS), which comprises two sectors, HGA and HGB. The gain difference between the HGS and MGS is about 2 orders of magnitude, while the overall HGS radiometric gain difference from the LGS is about 5 orders of magnitude. DNB gains are calibrated using 3 on-board calibrators (OBC), which utilize solar diffuser (SD), black-body (BB), and space view (SV) signal measurements to produce consistent long-term sets of calibration reference data for the EV observations.

This paper aims to present a review of the approach that the NASA VIISR Characterization Support Team (VCST) has taken with respect to calibrating and monitoring the trending behavior of the DNB gain responses and background offsets. This approach relies on the constantly collected OBC data from the SD, BB, and SV sectors, with continuous updates of the gain ratios and offset drifts over short- and long-term periods [7–9]. The crux of this approach is the collection of well-characterized LGS SD observations. During each solar calibration event, the cross-stage gain ratios are estimated using simultaneously observed SD signals. The background offset drift can then be determined using nighttime BB dark signals free from airglow contamination. A time-dependent throughput change as a function of wavelength is used to build a modulated relative spectral response (RSR), which is applied to calculate the DNB gains. To deal with stray light contamination in affected areas around the southern and northern terminators, a stray light estimation is performed, and its correction is implemented.

In this paper, Section 2 provides a brief review of VIIRS DNB operation and degradation behaviors, as well as calibration methods used by the VCST. Section 3 presents the DNB calibration performance to show the F-factor and gain ratio trending in the three gain stages. DNB dark offsets are presented in different aggregation modes and different gain stages. The SD derived gain is compared with the lunar-based gain trending. The signal-to-noise ratio (SNR) is also evaluated for all detectors and aggregation modes. Section 4 discusses the stray light correction strategy. Section 5 compares the DNB calibration algorithms used by VCST to those used by the NOAA Interface Data Processing Segment (IDPS), and discusses potential avenues for further improvement. Section 6 presents our concluding remarks.

## 2. DNB Description and Calibration

The DNB's four gain stages (low, middle, and two high sectors) are mapped onto four CCD detector arrays. These arrays comprise of 672 detectors, which are co-added during the on-board processing of the DNB observations and converted into 16 aggregated detectors along-track, with a similar swath size of about 750 m. In the along-scan direction there is another application of aggregated observations with a time-delay-integration (TDI) scheme. The detectors (sub-pixels) are grouped differently for each gain stage, with 250 for the HGS, 3 for the MGS, and 1 for the LGS. The gain sensitivity of the arrays is as follows: HGS $\sim \mathcal{O}(2) \times$ MGS; MGS $\sim \mathcal{O}(3) \times$ LGS.

Another aggregation scheme is applied to the processing of EV observations to get a per-pixel spatial resolution which is similar in size. The recorded gain responses from the EV sector are divided into 64 aggregation zones, which are split in pairs symmetrically about the nadir. This creates 32 pairs, which are designated as aggregation modes. As a result of this distribution and aggregation of spatial zones, together with the scan times of the detector arrays, the different modes are sampled on varying times, each mode's gain

response varies, and each mode should be calibrated separately. The LGS, for example, requires independent calibrations for each of the 32 aggregation modes, each of the 16 detectors, and each of the 2 HAM sides, resulting in 1024 calibration coefficients. The detail layout scheme of the aggregated modes is available in [9,10]. For the middle and high gains, an established approach is to add calibration coefficients for the gain ratios relative to the LGS, which also depend on each mode, detector, and HAM side, resulting in the same number of coefficients as for the low gain for each of two types of ratios. The VIIRS DNB observations are designed so that the SD, SV, and BB sectors cover a single aggregation mode for the adjoining scans, after which these move to cover the next mode, and so on for each pair of scans. In practice, the SD, SV, and BB sectors have 36 programmed spaces which are distributed between the EV 32 aggregated modes and 4 other spare modes, and are not included in the calibration schemes as they are designed for testing purposes.

Routine DNB calibrations need to consider a degradation effect, which depends on the measured wavelength and covers the relevant spectral range. This is a natural response of the VIIRS optical system (indeed any space-borne optical system) to in-situ processes that affect screen and sensor materials interacting with the space environment. Since this process is heterogeneous over wavelengths, it is also time-dependent. For the VIIRS DNB, this degradation is represented by the modulated function of the relative spectral response (RSR) [11]. Figure 1 shows the DNB RSR for different times during the mission (pre-launch, 2012, 2015, 2021). For the pre-launch period, the RSR is measured directly and follows the SNPP specifications. The other 3 time periods shown in Figure 1 are for the derived RSR values. Derivation of the on-orbit time-varying RSR is done by considering the reflective solar bands (RSB) as varying mainly due to the degradation effect. The RSB calibration coefficients for bands M4-M7 are then used to derive the RSR at a given measurement time. Since these RSB bands cover the wavelength range of 0.545 to 0.885 μm, it overlaps with the DNB wavelength range of 0.5 to 0.9 μm. As time progresses, there is a decrease of the RSR values at longer wavelengths, although these changes seem to be somewhat larger in the early mission times (see Figure 1). This is caused by the RTA's stronger degradation in the near-infrared regime. The 2015 and 2021 data values in Figure 1 show that there is a decrease in the degradation rate, as the RSR curves become closer to each other over the wavelength range.

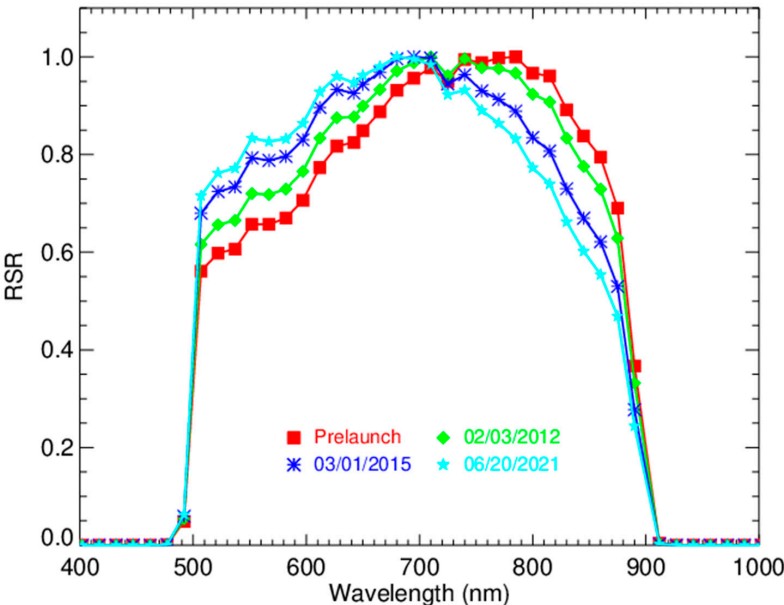

**Figure 1.** Pre-launch and on-orbit modulated DNB RSR. 3 February 2012, 1 March 2015, and 20 June 2021 present the first, the fifth and the tenth year in operation.

Figure 2 shows the VIIRS SD degradation as measured by the solar diffuser stability monitor (SDSM) [1,11–13] at five times during the mission: orbits of 2000, 5000, 15000, 35000 and 50000, which correspond to 17 March 2012, 14 October 2012, 19 September 2014, 30 July 2018, and 22 June 2021, respectively. The VIIRS has operated normally for about ten years since the nadir door was opened. About 42% degradation is observed at wavelength 0.412 μm so far. At this point in the on-orbit operations, the largest degradation is confirmed as being at the shortest wavelength [11–18]. In this figure, a coverage of the DNB wavelength is plotted background in green dots.

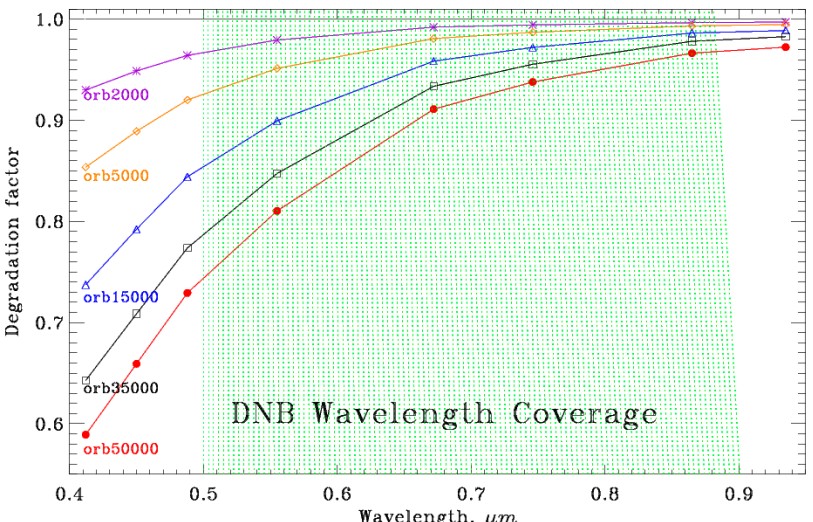

**Figure 2.** VIIRS SD degradation at five times during the mission as marked by the orbit numbers. Different colors denote different orbits. The DNB wavelength coverage is shaded in green.

Our DNB radiometric calibration uses SD observation during orbital solar calibration events, SD data outside these events, and night-time BB data to estimate the LGS gain, the cross-stage gain ratios, and dark offset drift (also designated as DN0), respectively. The time-dependent modulated RSR is applied to ensure LGS gain accuracy. Signal measurements from deep space scenes during a pitch maneuver near mission start, which represent constant reference points, are used to improve each gain's dark offset accuracy.

In a similar manner to the VIIRS RSB calibrations, the radiometric calibration coefficient of the DNB is termed the F-factor. This factor represents the inverse of the DNB's detector gain, and as such is defined for all DNB stages as F-LGS, F-MGS, and F-HGS. In general, the DNB top-of-atmosphere (TOA) radiance in the EV images is defined as

$$L_{DNB}^{TOA} = F_{gain} \cdot \frac{dn_{EV}}{RVS_{EV}} \tag{1}$$

where $F_{gain}$ is the gain-stage calibration factor, $RVS_{EV}$ is the response versus scan angle for the DNB at the EV image, and $dn_{EV}$ is the detector response at the EV scene after its background subtraction. The latter corresponds to a low, mid or high gain, based on the radiance level of each pixel.

Determining the F-factor value for the LGS relies on SD measurements during fully illuminated conditions. These conditions, which are colloquially designated the "sweet spot", are for Sun declination angles of 14° to 18°. The LGS is the only gain stage able to use the SD observations for on-orbit calibration, because the mid-gain and high-gain detectors are too sensitive and saturated when viewing the fully illuminated SD [7–9].

The LGS calibration is implemented by comparing the expected radiance values on the SD with the actual signals recorded from it (digital counts, $dn_{SD}$). This signal is background-subtracted using the SV recorded signal at the same time. The gain coefficients, or F-factors, which are then applied as calibration gain tables to the production of calibrated radiance in

Level 1B (L1B) products, are then defined by $L_{SD} = F \cdot dn_{SD}$, where the DNB SD radiance is computed as

$$L_{SD} = \frac{cos\theta_{SD} \cdot RVS_{SD} \cdot \int BRDF(\lambda)\tau_{sds}H(\lambda, t)RSR(\lambda, t)\Phi(\lambda)d\lambda}{4\pi d_{SD-SUN}^2}. \tag{2}$$

where $\theta_{SD}$ denotes the angle between the normal of the SD-to-Sun vector and the SD surface; $\Phi(\lambda)$ presents the solar spectrum distribution at a distance of one astronomical unit; $d_{SD-SUN}$ denotes the Sun-to-spacecraft distance in astronomical units; $\tau_{sds}$ is the transmittance of the SD pinhole screen; $H(\lambda, t)$ presents the SD degradation factor, which is corrected by the lunar calibration results in practice. It is estimated by an integration of the DNB RSR, $RSR(\lambda, t)$, and the RSB-derived SDSM H-factors; $RVS_{SD}$ denotes the response versus scan angle of the HAM at the SD view center; $BRDF(\lambda)$ denotes the bidirectional reflectance factor of the SD, and was estimated pre-launch using reference samples based on the National Institute of Standards and Technology (NIST) reflectance standards.

The MGS and HGS F-factors are not calculated directly, but utilize a formulation based on the gain ratios, so that

$$F_{MGS} = F_{LGS} \cdot \left(\frac{dn_{LGS}}{dn_{MGS}}\right) \tag{3}$$

$$F_{HGS} = F_{LGS} \cdot \left(\frac{dn_{MGS}}{dn_{HGS}}\right) \cdot \left(\frac{dn_{LGS}}{dn_{MGS}}\right), \tag{4}$$

where the *dn* values for the first ratio of LGS/MGS are carefully selected to have mid-gain values below the saturation and low-gain values in corresponding scans. The MGS/HGS ratio values are selected to have HGS values below the saturation and MGS values in corresponding scans. Hence, the HGS and MGS gain values are determined under partially illuminated SD conditions (i.e., scans outside of the sweet spot). The HGS and MGS F-factors are calculated based on the calibrated LGS F-factor results and the estimated gain ratios.

Overall, the DNB calibration formulation is computed for each combination of gain stage, detector, aggregation mode, and HAM side. The LGS gain, gain ratios, and dark offset values are recorded as a time-series and saved as historical data sets. A 1-day smoothing window is applied to each historical data set to reduce random noise to produce the calibration and prediction look-up-tables (LUT).

## 3. Calibration Performance

As presented in the previous section, the LGS gain is estimated using SD measurement. This is done under the conditions at which the SD is fully illuminated by the Sun, while the incident light is attenuated by the SD screen. These conditions are set by defining a sweet spot of declination angles in the range of 14°–18° and azimuth angles in the range of 14°–44.8°. Since both the middle and high detector arrays have a high sensitivity of gain response, they will saturate under such conditions. Hence, the MGS and HGS gain are not calibrated directly in this approach. However, we can select unsaturated data to calculate the MGS/LGS and HGS/MGS ratios. These gain ratios are calculated by using scans outside the sweet spot, where the detectors are only partially illuminated and the gain response level is low enough to record an unsaturated signal. Therefore, by applying the time dependent modulated RSR, we can perform DNB LGS calibration in each SD calibration event.

Figure 3 plots the DNB LGS F-factor coefficients in units of W/(cm²sr). In this figure, the left chart presents the daily averaged mode-1 (nadir), and the right presents mode-15, which is centered at ±24.5°. By normalizing at the beginning data point, all DNB detectors show similar trends in these cases. As the VCST uses the SD *dn* signals to calculate the gain ratios between the DNB's middle-to-low and high-to-middle gain stages, including scans outside the sweet-spot. Figure 4 shows the DNB gain ratio trend of middle-to-low from aggregation mode-1 and mode-15. The case of mode-1 presents relatively stable trending,

however higher modes experience larger fluctuations during ten years of operation for all detectors, as expected.

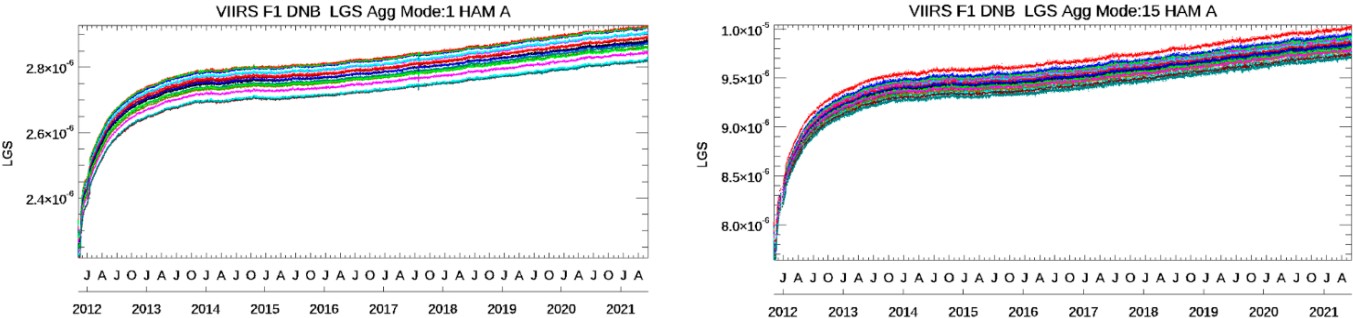

**Figure 3.** DNB daily averaged LGS F-factor trending at two aggregation modes (mode-1 and mode-15). Different colors are used to plot the DNB 16 detectors.

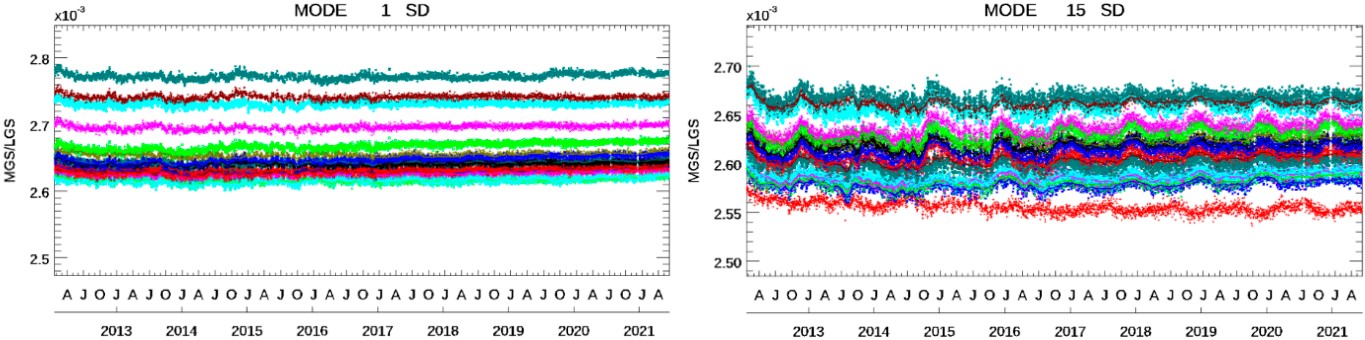

**Figure 4.** Ratio trend of DNB middle-to-low using SD *dn* signals at two aggregation modes (mode-1 and mode-15). Different colors are used to plot the DNB 16 detectors.

Figure 5 shows the DNB high-to-middle ratio trend from aggregation mode-1 and mode-15. Using Figures 4 and 5, together with LGS F-factors in Figure 3, the DNB MGS and HGA gain trending can be obtained in the daily averaged aggregation modes specified. Similarly, we can obtain the MGS and HGB gain ratio trending. By averaging them, the HGS over MGS ratio can be determined.

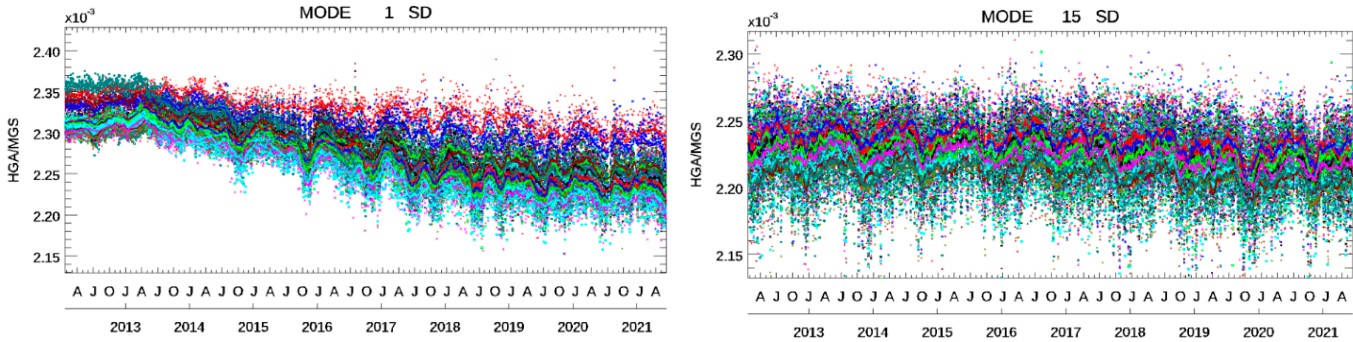

**Figure 5.** Ratio trend of DNB high-to-middle using SD *dn* signals at two aggregation modes (mode-1 and mode-15). Different colors are used to plot the DNB 16 detectors.

The ratios of these higher gains (HGA/MGS and HGB/MGS) show much stronger fluctuations than the MGS/LGS ratio, which is expected as they are driven by the higher sensitivity of the HGS detectors. Figure 5 also shows that there is a long-term (multi-year) trend that differs between the different modes. Aggregation mode 1 (Figure 5 left) has a decreasing trend of the gain ratio for all detectors, while aggregation mode 15 (Figure 5 right) has a relatively flat behavior among the 16 detectors. A similar downward trend

can be noticed for aggregation modes less than 10. This behavior is still under analysis, but it can be noted that the downward trend indicates a difference in the degradation rate between the high and middle gains.

The computation of the LGS gain requires a reference value, so that the on-board measured SD signal responses can be calibrated properly by removing what is referred to as dark offset. The EV dark offset depends on the sample view and the recorded signals are aggregated through different modes. Hence, it cannot be used as a direct calculation from the calibration view data through the SV sector measurements. The VCST DNB calibration approach employs a more accurate estimator for the dark offset by utilizing the BB signals, which effectively removes the need to consider the airglow signal contamination that exists for the SV and EV sectors. By conducting a special pitch maneuver to include corrections of the reference offset drift for the EV, it has been demonstrated that it is the same as the offset drifts measured for the LGS SD.

As for the LGS gain and the gain ratios, the dark offset calculation depends on mode, detector, and HAM-side. This is because the field-of-view (FOV) is not constant between the modes. Maintaining a near-constant spatial resolution on the ground means that mode 1 (closest to nadir) and mode 32 (closest to edge of scan) cover the largest and smallest FOC, respectively. Figure 6 shows the trending of the dark offset signals for the LGS (left) and MGS (right) for mode 1, HAM-side A, and all 16 detectors. The trending of the dark offsets is based on minimal values from the OBC data of the SD, SV, and BB. Pitch maneuver values on February 20, 2012 are then used for normalizations.

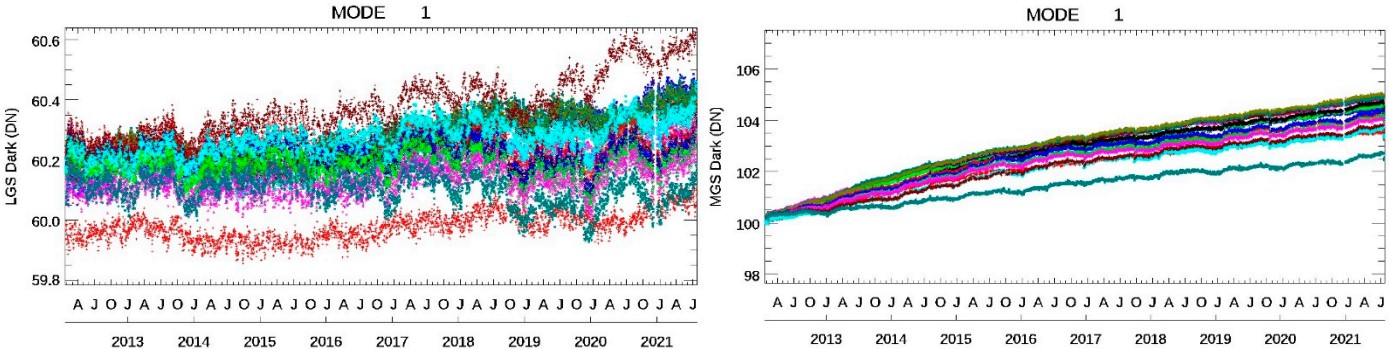

**Figure 6.** DNB dark signals of Mode-1 in LGS and MGS. Different colors are used to plot the DNB 16 detectors.

Figure 7 shows the DNB dark signals of both HGS-A and HGS-B at HAM-1 in mode-1 and mode-15. As the flight software FSW_0x4018 was uploaded on 8 January 2020, its impacts result in a large change in the case of HGS-B. The magnitude of the changes associated with this software update are mode-dependent but confined to HGS-B (HGS-A is unaffected). A gradual increase is observed for the HGS and MGS dark offsets, however, changes to the MGS dark offsets are very small and less than 1 digital count. The changes of the LGS dark offsets are within 0.5 digital count. In all, the trending of the dark signals is nearly flat for all 32 aggregation modes in the LGS and MGS, but lower modes (less than 10) in the HGS show an upward trend with up-down fluctuations due to new-moon and full-moon times each month. However, the HGS dark signals have relatively flat trends for four high modes of modes-29, 30, 31 and 32.

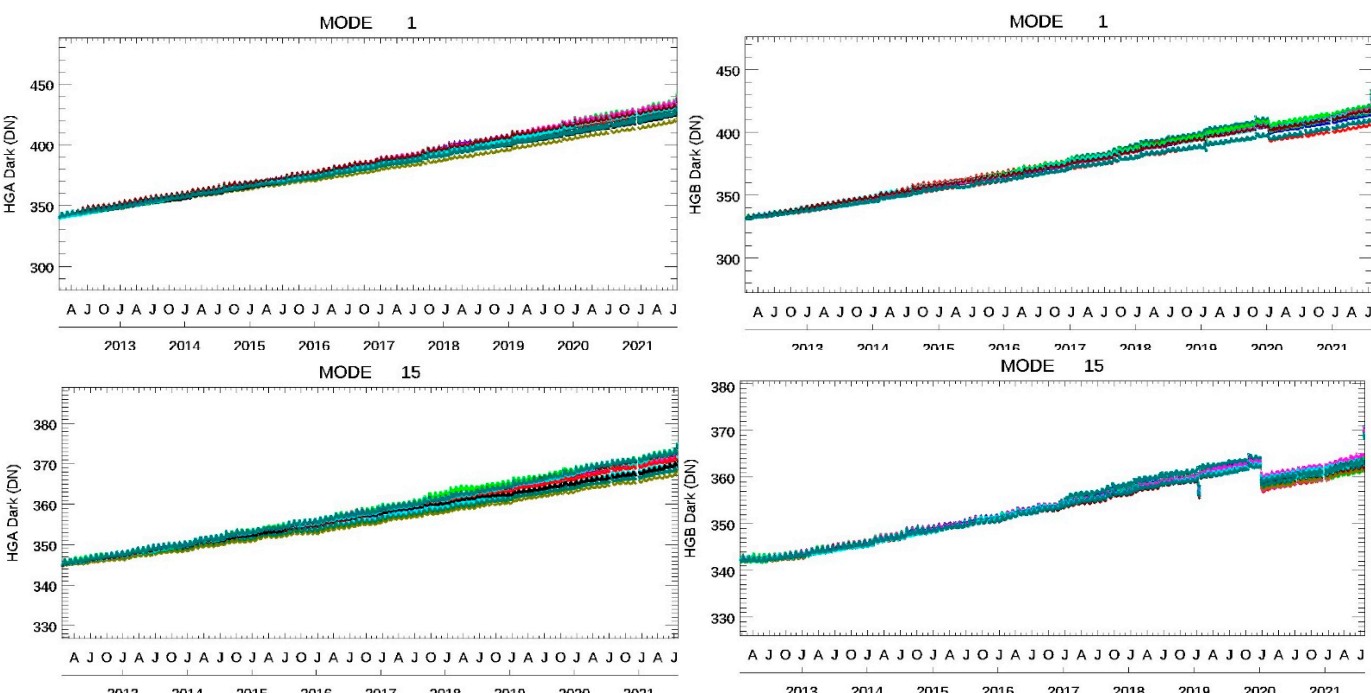

**Figure 7.** DNB dark signals of HGS-A and HGS-B in mode-1 and mode-15. Different colors are used to plot the DNB 16 detectors.

There are signal-to-noise ratio (SNR) requirements at the minimal radiance $2.5 \times 10^{-9}$ Wcm$^{-2}$ sr$^{-1}$ in the S-NPP DNB system design specification, such that (1) SNR is larger than 6 as modes are less than or equal to 21; (2) SNR is larger than 5 for the other modes. This specification applies in the HGS stage only. Figure 8 shows all 16-detector SNRs at the BB for mode-1 and mode-15. A similar behavior is observed in all other modes. As the S-NPP VIIRS is an aging instrument on-orbit, the calculated SNR is expected to decrease with time but shows no signs of violating the specifications.

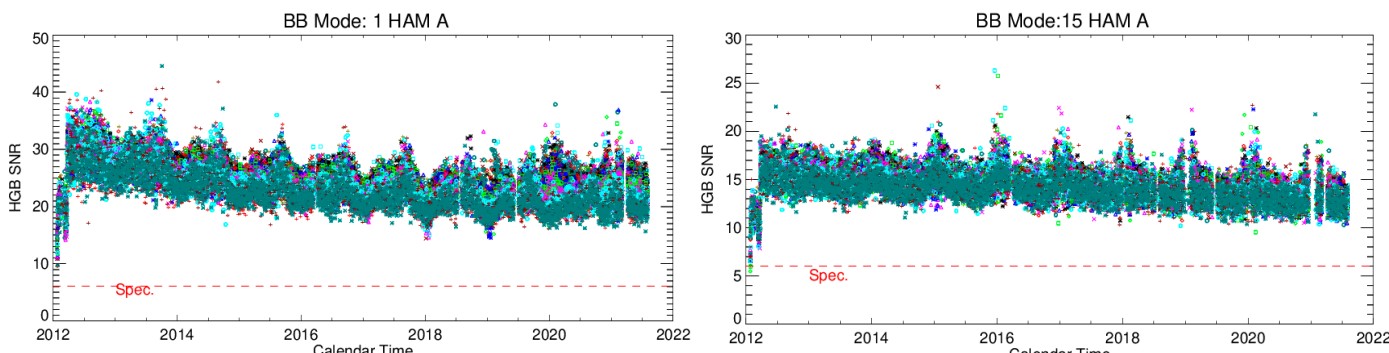

**Figure 8.** DNB SNR trend for HGS-B in two aggregation modes (mode-1 and mode-15). Different colors are used to plot the DNB 16 detectors.

Figure 9 shows the most recent three-month averaged SNR (Mar-21/Apr-21/May-21) in different modes. We confirm that higher modes have a smaller SNR, as expected. The averaged results illustrate that the S-NPP DNB is above system specifications for all of modes 1–32.

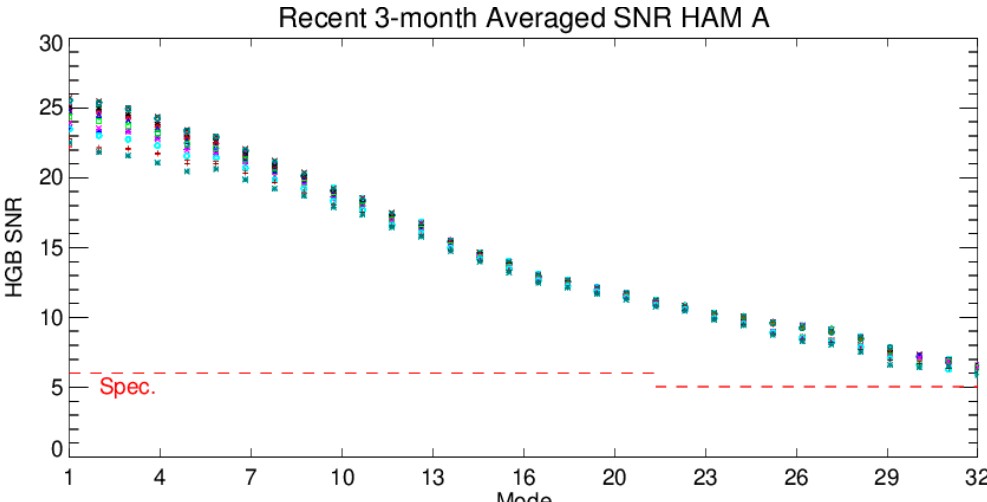

**Figure 9.** Recent 3-month averaged SNR versus Modes in HGS-B for HAM-1. Different colors are used to plot the DNB 16 detectors.

## 4. Stray Light Effect and Correction

Stray light in the DNB night images has been observed since the beginning of the VIIRS S-NPP mission. It happens in about 25% of the night scenes, covering both the northern and the southern hemispheres [10,14–16]. The maximum of the contamination reaches a level of $5.0 \times 10^{-9}$ Wcm$^{-2}$ sr$^{-1}$, while the radiance from the meaningful earth surface objects can be as low as $1.0 \times 10^{-9}$ Wcm$^{-2}$ sr$^{-1}$. Thus, the quality of the earth view images is severely reduced.

In the northern hemisphere, the stray light starts when the spacecraft's solar zenith angle is greater than 90 degrees. It gradually increases until the spacecraft goes into the shadow of Earth. During this period, sunlight shines on the facet of the EV opening. In the southern hemisphere, the same source exists, but the time is reversed. In addition, sun light can enter via the SD port which creates a second layer of stray light in the southern hemisphere data.

Figure 10 illustrates the stray light correction method in a particular data range termed as bin-6. Three detectors are presented in three different colors. In DNB night images, each scan has 4064 pixels, which are separated into 127 bins dependent on each HAM side and detector. Along with the solar zenith angles (SZA), a baseline is established using the lowest portion of radiance in the area SZA > 119°. Each bin has 32 pixels, and the stray light estimation is based on these 32 pixels, where the lowest 20% of values are used as the data source for stray light estimation. The corrected signals are the results of the raw signals, subtracting the stray light estimates and adding back the baseline values. The stray light estimates are written in a LUT, which is dependent on SZA, scan angle, detectors, HAM sides, and hemispheres.

Stray light correction methods have been developed. The correction LUTs are generated and applied mission-wide during NASA L1B reprocessing. The IDPS forward calibration is performed using on-orbit *AutoCal*. As the stray light profiles experience a repeatable pattern during the yearly Earth-Sun-spacecraft geometric cycle, a stray light estimate can be determined from exist LUTs generated in the previous years.

The same principles are used in both NASA VCST and NOAA IDPS for developing the DNB stray light estimation algorithm [7,8,14–17]. During a new moon, signals observed over the dark Earth scene are the result of a combination of stray light and airglow. There are two major differences between the IDPS and VCST approaches. One is that the IDPS method uses dark Earth scene signals, and an Earth nighttime light map is applied to exclude nightlights during a new moon period. However, pixels with nightlights are treated as outliers in the VCST approach. Nightlights are only on a tiny fraction of the Earth's surface. A small outlier exclusion threshold (say 20%) may effectively remove

most of the non-straylight pixels. The second difference is that the VCST calculates the dark scene signals via smoothing over the spacecraft zenith and scan angles. However, the IDPS directly averages the dark scene signals [9,16,17]. In practice, the VCST uses neighbor correlation effects to decrease estimation uncertainty on dark scene signals. Using a smoothing function to filter out measurement noise and distortions, more accurate dark signal estimates can then be obtained.

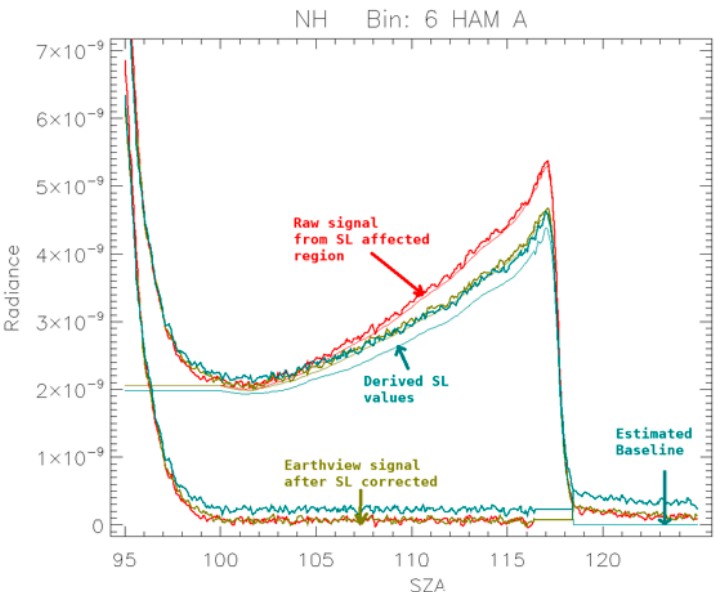

**Figure 10.** An illustration of stray light correction in bin-6: three detectors are presented in different colors. For each detector, the top noisy curve represents its raw signal. The smooth curve underneath the raw signal is its stray light estimate. After stray light correction, the signal is shown at the bottom.

Figure 11 shows a DNB image example over the Mediterranean before and after stray light corrections. A Google GEO-location map of Europe is shown on the left of this figure. On the middle, the DNB night image at 1:40 GMT, 11 April 2021, is presented, where the stray light impacts the upper part of the nighttime image. After stray light correction, the night image is on the right of Figure 11. Without any noticeable degradation in its image quality, the stray light contamination was effectively removed in this night image.

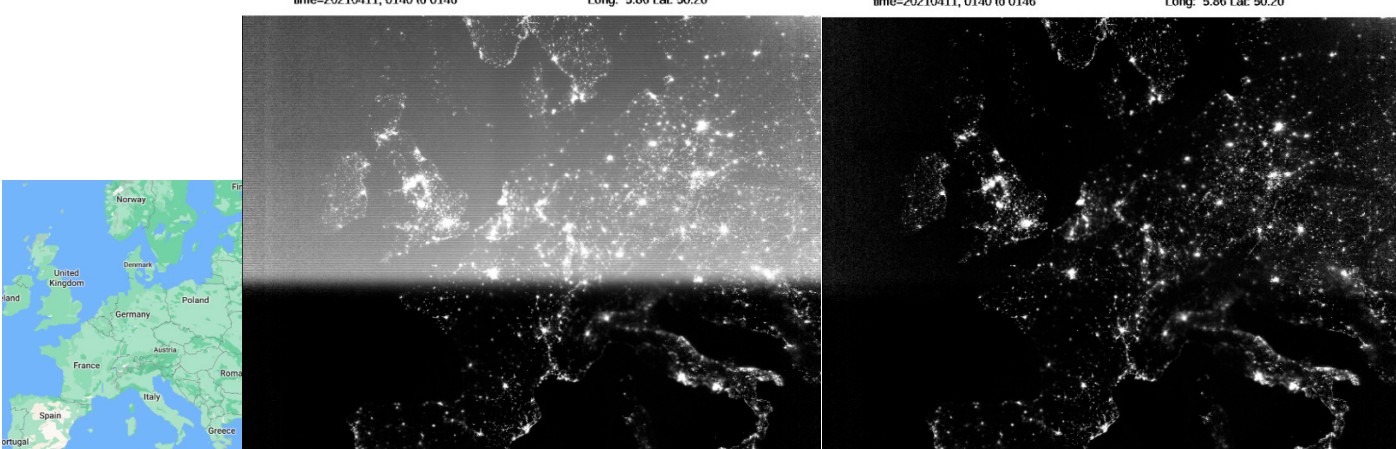

**Figure 11.** SNPP DNB images from 11 April 2021, 1:40 GMT. Images before/after stray light correction are located middle and right, respectively.

## 5. Discussions and Future Work

The first Joint Polar Satellite System (JPSS-1) was launched on November 18, 2017 and renamed NOAA-20 after launch. It has joined the SNPP satellite in the same orbit and operates about 50 min ahead of SNPP so as to have an area of overlap in observational coverage. It is valuable to compare their early mission behaviors.

Figure 12 shows the SNPP and NOAA-20 VIIRS DNB LGS F-factor trends. Focusing on the first three-years of on-orbit operations, NOAA-20 illustrates extremely stable performance for all detectors, modes, and HAM sides. However, the SNPP LGS F-factor shows a large change in the early mission [15,18–25].

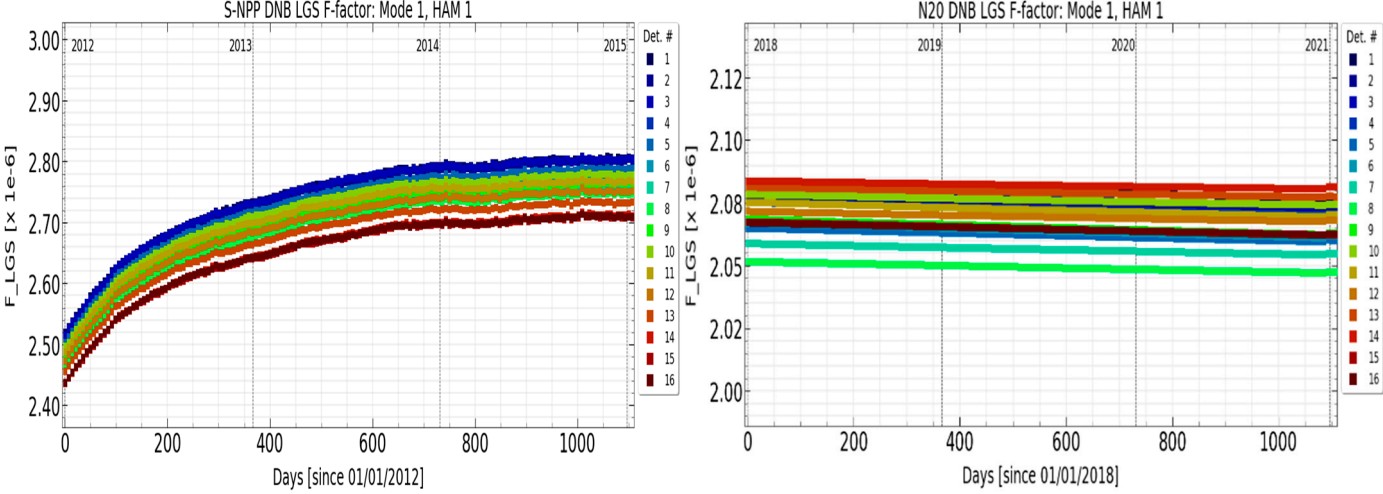

**Figure 12.** SNPP and NOAA-20 VIIRS DNB F-factor comparisons in the case of LGS mode-1 at HAM-1. DNB 16 detectors are plotted in different colors.

Another area of comparison can be investigating different calibration methods. One such method is conducted by the VCST, and is presented in this paper. It is used in the NASA Land PEATE and/or L1B at the site of the atmosphere archive and distribution system (LAADS) (http://ladsweb.nascom.nasa.gov, accessed on 6 October 2021). Another one is performed by the IDPS, which is officially employed for the NOAA SDR, downloaded from (https://www.ngdc.noaa.gov/mgg/gravity/, accessed on 6 October 2021). In the IDPS method, two VIIRS Recommended Operating Procedures (VROPs) are performed to determine the gain ratios (VROP705) and dark offsets (VROP702). There are several updates of the dark offset calibration since early mission. Currently, they are determined based on analyzing the VROP data as well as the onboard blackbody data [6,16,17]. Figure 13 shows the SNPP F-factor comparisons using VCST and IDPS calibration methods [17,22]. The discontinued IDPS DNB LGS F factor shown in Figure 13 also reflects how the IDPS LGS gain calibration has been updated since the beginning of the mission. The continuous DNB LGS F factor was used in the reprocessed NOAA NPP VIIRS DNB SDRs. The details of the reprocessing are available [6,17]. The case of mode-1 and HAM-1 was used. Similar trend results were observed for other modes and HAM-2. Before the middle of the year 2013, there was a large difference (more than 10%) between the VCST and IDPS, where the pre-launch RSR was applied. Using the modulated RSR in the IDPS after May 2013, both methods agree very well. In April 2014, large changes were seen in the IDPS as Sun eclipse data were involved in calculations. Also, an H-factor jump occurred in Feb 2019. It is necessary to mention that the IDPS started an *Auto-cal* processing after January 2016, which is marked by a dash vertical line in Figure 13 [26–28]. A package of monthly calibration LUTs was delivered by the IDPS before that time. In the *Auto-cal* processing, on-orbit data are available for instant calculations. In the bottom of Figure 14, a comparison difference in percent is plotted. The majority difference between the VCST and the IDPS was less than 1.0% for all 16 detectors.

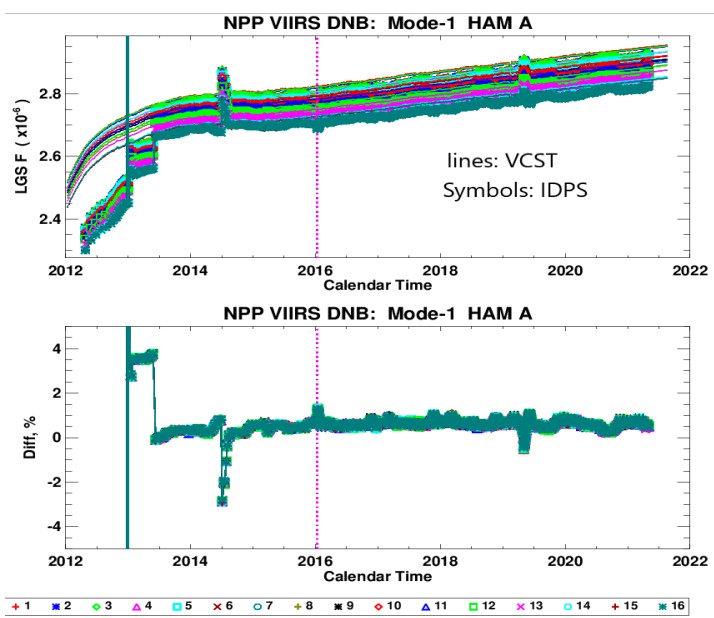

**Figure 13.** VCST and IDPS SNPP DNB F-factor comparisons in the case of LGS mode-1 at HAM-1. DNB 16 detectors are plotted in different symbols/colors.

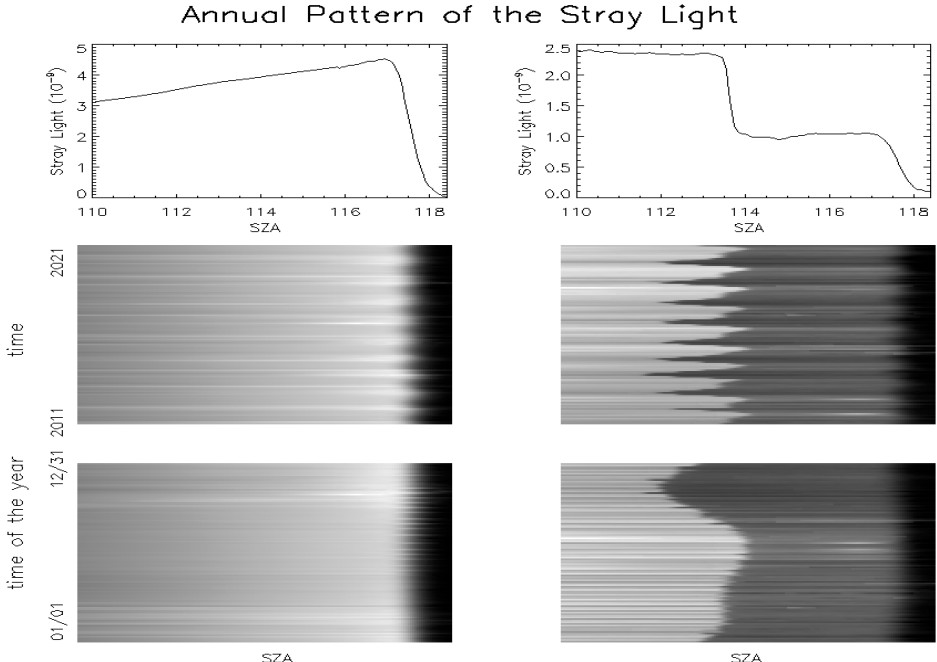

**Figure 14.** SNPP VIIRS DNB stray light patterns in 10 years. Left charts are examples from the northern hemisphere, and right charts are examples from the southern hemisphere.

The DNB stray light estimation and correction algorithm that VCST developed has been successfully implemented and considerably reduces the light contamination in Earth images from both the northern and southern hemispheres. Stray light estimates are time-dependent. These are obtained for every new moon event, but also depend on the specific geometry between the spacecraft and the Sun. Figure 14 shows DNB stray light measurement patterns for the 10-year mission. The top two charts show examples of stray light estimates in one selected bin at the northern hemisphere and the southern hemisphere, respectively. The middle charts are the corresponding stray light estimate contours over 10 years. The bottom two charts are zoomed contours for one year. In Figure 14, notice

that the stray light shifts along the solar zenith angle (SZA) with time. When the stray light estimate LUT is applied to the Earth view scenes at times that differ from when the LUT is obtained, it leads to an over- or under-correction of the radiance. This is most noticeable around the peak of the stray light, where a sudden drop occurs [10,16,29]. We have developed a method to correct for this effect by inserting additional stray light LUTs between new moon dates. This is achieved by analyzing repeatable pattern stray light contamination over an annual cycle. When we apply this enhancement correction to the raw radiance in the affected region, we see that most of the light contamination impacts are decreased substantially. The obvious drawback of this approach is the calculation and maintenance of a large data volume, especially if this correction enhancement would be maintained for the entire mission.

We note that there is room to improve upon the estimation and correction approaches for the S-NPP DNB stray light contamination. One option, which requires a deeper dedicated analysis, is to convert the correction estimate LUTs into parameterized tables. By using parameters in the estimate LUTs, the volume of the LUTs can be significantly reduced without compromising the image quality. Since the stray light has shown a robust, repeatable pattern, we can plan analysis work that produces specific expressions to be used to define variations at a particular time of the year. The expected LUT volume might be only 1/10 of what we currently have. With such a parameterized approach, there will have to be an additional step of parameter derivation and update every certain period, to maintain the validity of the derived expressions. However, these will not have the same cadence and data analysis burden of the approaches currently employed to derive the detailed stray light correction LUTs.

## 6. Conclusions

In this paper we presented a review of the background, methodology, and performance of the DNB, which is part of the SNPP VIIRS instrument Earth observation portfolio. The task of on-orbit gain trending analysis and calculation of calibration factors has been continuously monitored, and its algorithm schemes have been continuously improved and validated by the NASA VCST. Over the 10-year mission length, the calibration performance has been well-characterized and robust, especially with respect to DNB degradation and calculation of dark offset trending. Both the gain and offset show stable long-term trends which are characterized per aggregation mode, detector, and HAM-side. Over time, all gain stages (low, middle, high) show a gradual decrease, which is driven by the LGS gain response, as the gain ratios seem to remain stable. While stable over mission-long trending, the HGS gain response exhibits large fluctuations, with amplitudes varying for the different modes. The high sensitivity of this stage, together with low SNR, is the main reason for this behavior, while its calibration scheme includes two conversions of signal utilizing the MGS/LGS and HGS/MGS gain ratios. Consideration of the DNB stray light corrections to be applied to EV scenes involves calibration of mode, detector, and HAM-side dependent data, all at high sensitivity levels. This is mostly due to the contributions of HGS and MGS to this light contamination. The discussion of stray light correction in this paper has shown that the VCST-applied algorithm demonstrates an effective and robust approach eliminating image artifacts due to light contamination. In general, the VCST and IDPS results for LGS gain calibration agree well with each other, with recent results having a discrepancy of up to 1% for all modes, detectors, and HAM-sides.

**Author Contributions:** Conceptualization, H.C., J.S. and X.X.; data curation, C.S., G.S. and H.C.; formal analysis, C.S., G.S., H.C., J.S.; methodology, C.S., G.S., H.C., J.S. and X.X.; writing—original draft, H.C.; writing—review and editing, G.S. and H.C. All authors have read and agreed to the published version of the manuscript.

**Funding:** This work received no external funding.

**Institutional Review Board Statement:** Not applicable.

**Informed Consent Statement:** Not applicable.

**Acknowledgments:** The authors would like to thank other members of the VCST for their technical discussions and assistance, especially Kevin Twedt, Daniel Links and Amit Angal for their helpful comments.

**Conflicts of Interest:** The authors declare no conflict of interest.

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
