# Peer review of "SNPP VIIRS Day Night Band: Ten Years of On-Orbit Calibration and Performance"

_remotesensing, doi:10.3390/rs13204179_

Round 1

Author Response

Thanks for this reviewer's comments. All comments are accepted. The point-by-point responses are listed below:

1) Yes. corrected and delete the sentence.

2) Yes. "change" is used to replace "drop"

3) Yes. Sentences are added to reflect reviewer's suggestions, and the mentioned reference is inserted as a new reference paper.

4) Based on the reviewer's comments, all quoted references are re-arranged/corrected. The original ref.13-19 are deleted. In our initial paper, we discussed some lunar validation results, but removed as the final submitted version, however forgot to remove relative references.

Reviewer 2 Report

  • figure 1: quality of the print is not very high, could be improved
  • please check all figures, some of them appear unclear/deformed in the paper
  • figure 11 : i would remove or separate the map and mention 'location Europe', this makes the image visually more logic
  • on the SD calibration : degradation of the DNB is observed, but was degradation on the solar diffuser monitored as well ? please mention, somewhere
  • page 5 : mode-1 is the nadir, for mode 15 I didn't find an explanation (view angle?) -> i might have missed it, so make shore it's there. page 3 gives a description-> it might help if you provide also a scheme for this aggregated layout
  • i would move the comparison between snpp and noaa-20 outside the discussion section. the discussion should in my view elaborate on the findings of the above sections and explain more the instrument behaviour and calibration findings. you just present both graphs (figure12) and do not add an explanation to the different behaviour (naybe there is none?)
  • conclusions are fine

Author Response

Thanks for this reviewer's comments. The point-by-point responses are listed below:

1) Yes. Figure 1 is re-plotted using a high resolution. Other charts are checked fine with appearance in this paper.

2) As suggested, "Europe" is inserted in the explanation text. After careful considerations, we plan to keep the snap chart as it is. It may help to provide a visual identification to the night images. Yes. On the SD calibration, degradation of the DNB is observed. The degradation of SD is monitored as well. Figure 2 shows it at five times during the mission marked by the orbit numbers. The DNB wavelength coverage is shaded in green.

3) Yes. Mode-15 is centered at +/- 24.5 deg., which is inserted in the explanation text. To provide details on mode layout, a sentence "The detail layout scheme of the aggregated modes is available in [9,21]." is inserted in the description line-105.

4) We have accept the other reviewer's suggestions, and insert some updated information/explanation on IDPS results. Thus we keep there as it is. Figure-12 is to show the difference in early mission between SNPP and N20 DNB. There might be a system design improvement for N20 DNB, however, we don't have detail information at this moment.

Reviewer 3 Report

Written in an eloquent manner, the manuscript is full of useful information. The effects of stray light contamination on the DNB and how to rectify them are discussed. Performance validations are provided in the form of comparisons to the calibration techniques used by NOAA's operational Interface Data Processing Segment to determine system performance. It is also debated whether or not to continue with stray light adjustments.

Author Response

Thanks for this reviewer's comments. 

Round 2

Reviewer 2 Report

paper is fine for publications, adaptations are fine

This manuscript is a resubmission of an earlier submission. The following is a list of the peer review reports and author responses from that submission.